# The Octopus Trap of Takotsubo and Stroke: Genetics, Biomarkers and Clinical Management

**DOI:** 10.3390/jpm12081244

**Published:** 2022-07-29

**Authors:** Isabella Canavero, Nicola Rifino, Maurizio Bussotti, Tatiana Carrozzini, Antonella Potenza, Gemma Gorla, Giuliana Pollaci, Benedetta Storti, Eugenio Agostino Parati, Laura Gatti, Anna Bersano

**Affiliations:** 1Cerebrovascular Unit, Fondazione IRCCS Istituto Neurologico Carlo Besta, 20133 Milan, Italy; isabella.canavero@istituto-besta.it (I.C.); nicola.rifino@istituto-besta.it (N.R.); tatiana.carrozzini@istituto-besta.it (T.C.); antonella.potenza@istituto-besta.it (A.P.); gemma.gorla@istituto-besta.it (G.G.); giuliana.pollaci@istituto-besta.it (G.P.); benedetta.storti@istituto-besta.it (B.S.); laura.gatti@istituto-besta.it (L.G.); 2Cardiology Rehabilitative Unit, IRCCS Istituti Clinici Scientifici Maugeri Clinical Scientific Institutes, 20138 Milan, Italy; maurizio.bussotti@icsmaugeri.it; 3Neurology Rehabilitative Unit, IRCCS Istituti Clinici Scientifici Maugeri Clinical Scientific Institutes, 20138 Milan, Italy; eugenio.parati@icsmaugeri.it

**Keywords:** Takotsubo cardiomyopathy, stroke, acute stress, genetic susceptibility, biomarkers

## Abstract

Takotsubo cardiomyopathy (TC) is a reversible cardiomyopathy mimicking an acute coronary syndrome, usually observed in response to acute stress situations. The association between acute ischemic stroke and TC is already known, since it has been previously reported that ischemic stroke can be both a consequence and a potential cause of TC. However, the precise pathophysiological mechanism linking the two conditions is still poorly understood. The aim of our review is to expand insights regarding the genetic susceptibility and available specific biomarkers of TC and to investigate the clinical profile and outcomes of patients with TC and stroke. Since evidence and trials on TC and stroke are currently lacking, this paper aims to fill a substantial gap in the literature about the relationship between these pathologies.

## 1. Introduction

Takotsubo is a Japanese word defining a specific trap for octopuses that became medically famous since it shares the shape with a cardiologic disease featured by a characteristic “apical ballooning”, which after that has been renamed Takotsubo Cardiomyopathy (TC) [1,2,3]. TC is a reversible cardiomyopathy mimicking an acute coronary syndrome, usually observed in response to acute stress situations. Other expressions referring to this condition include: “stress induced cardiomyopathy” [4], “broken heart syndrome” [5], “myocardial stunning” [6].

TC is a matter of interest for cardiologists but also for neurologists. TC is believed to be caused by an overshooting sympathetic response on the heart, thus being potentially triggered by many clinical scenarios that are associated with a major involvement of the autonomic nervous system. Among these, brain damage has also been described as a possible trigger for TC. Indeed, the combination of “broken heart” and “broken brain” turns out to occur in several pathological conditions. A quite robust association between TC and ischemic stroke is undeniable [7]. In addition, scattered literature describes the possible connivance of TC with other acute neurological disorders such as transient global amnesia [8], aneurysmal subarachnoid hemorrhage (aSAH) [9], epileptic seizures [10]. The true incidence of TC in association with these diseases is unknown. Some studies suggest that TC is present in 0.1–15% of aSAHs [11,12], others in a smaller percentage [13,14]. The prognosis of patients with aSAH and TC is mainly determined by the neurological damage, and evidence suggests not to postpone treatment of a ruptured cerebral aneurysm due to concurrent TC [15,16]. Nevertheless, whether TC worsens the prognosis in patients with other concomitant neurological disease, including stroke, remains to be clarified.

The pathophysiological mechanisms linking stroke and TC are still poorly characterized. The main unsolved question in the association between TC and stroke is determining which came first, and consequently, what would be the best management strategy to prevent either possible cardiac or cerebrovascular complications and implement patients’ outcomes. In fact, both cases of stroke after TC and TC after stroke are described [7]. However, another interesting perspective suggests that they can be both consequences of the same stressor. Indirect evidence provided by many clinical reports suggest that these two conditions sometimes have to be considered as appearing together rather than one determining the other, in the context of systemic acute stress responses [17].

## 2. Cardiological Assessment and Diagnosis of Takotsubo Cardiomyopathy

### 2.1. Symptoms and Signs

From a cardiological point of view, TC still has many dark sides and constitutes an important challenge both for diagnosis and treatment for the absence of a clinical hallmark. In fact, TC is a transient cardiac syndrome whose clinical features are similar to an acute coronary syndrome (ACS). During the initial evaluation in the emergency room the most common symptoms complained about by patients are chest pain, dyspnea, and rarely palpitations or syncope. Seldom, TC manifests itself with symptoms and signs arising from its complications, e.g., cardiogenic shock [18,19,20,21]. However, at the beginning, TC and ACS are substantially indistinguishable. 

### 2.2. Predisposition and Risk Factors 

Usually, medical history of TC patients lacks the most common risk factors for atherosclerosis, and in two-thirds of them a close prior emotional or physical distressful experience is reported. Curiously, almost 90% of TC cases occur in postmenopausal women [22]; among men, those undergoing chemotherapy are mainly affected [23].

Moreover, differently from an ACS that commonly emerges during the morning hours, TC is often observed in the late hours of the day, when the growing burden of the above mentioned stressful triggers is likely to concur in its development [24]. The preponderance of postmenopausal females suggests a hormonal influence, such as declining estrogen levels [25]. Vasomotor tone may be influenced by estrogens via up-regulation of endothelial NO synthase [26]. Furthermore, they can attenuate catecholamine-mediated vasoconstriction and decrease the sympathetic response to stress in perimenopausal women [27]. However, clear evidence demonstrating a correlation between hormones and the development of TC is lacking so far. Of note, in patients with TC, a high prevalence of psychiatric and neurologic disorders has been described. In fact, 42% had a psychiatric disorder (depression in half of patients) and 27% an acute, chronic, or former history of neurological disorders [28]. 

Patients with TC appear to have high prevalence of depression, anxiety, and type-D personality, which has been associated with an increased cardiovascular risk [29]. Increased norepinephrine response to emotional stress has been reported in depressed patients [30]. Similarly, an impairment of norepinephrine reuptake transporters has been described among patients with anxiety and panic disorders [31], and increased local levels of catecholamines, as seen in patients on antidepressants (e.g., selective norepinephrine reuptake inhibitors), may facilitate myocardial stunning [32]. 

Moreover, the regions of the insula or posterior fossa are mainly involved in patients with ischemic stroke or epileptic events, with a concurrent TC. It has been demonstrated that TC occurs after neurologic disorders, such as stroke, seizures, and subarachnoid hemorrhage, suggesting that these conditions may serve as predisposing factors for the development of TC [33]. 

Finally, a genetic predisposition to TC has been hypothesized, and it will be extensively discussed in Section 5.

### 2.3. Diagnostic Assessment

Diagnosis of TC is often challenging. Its clinical features may closely resemble ACS.

Upon admission, both ECG abnormalities (i.e., ST-segment elevation, T-wave inversion) and a raise of cardiac enzyme levels are common, but their magnitude is often relatively less evident than what is usually observed during myocardial infarction [3]. 

However, when an ACS is suspected in the acute phase, coronary angiography with left ventriculography is considered the gold standard diagnostic tool to exclude or confirm TC. In TC patients, it usually shows the lack of significant atherosclerotic disease. However, in 10–29% of cases, concomitant coronary artery disease is reported [28]. For these patients, the differential diagnosis can be even more challenging. 

Transthoracic echocardiography provides a quick assessment of the left ventricular (LV) wall motion abnormalities, in particular hypokinesia or akinesia of the midapical segments [24], while the cardiac catheterization, usually performed in conditions suggestive for an acute coronary syndrome, evaluates the anatomy of epicardial vessels [20]. Coronary angiography may alternatively show either completely normal aspect or noncritical stenosis of the coronary arteries, while the left ventriculography is appropriate both to make evident the pathognomonic wall motion abnormalities and to estimate the ejection fraction of the left ventricle [20].

Furthermore, a significantly lower left ventricle ejection fraction (LVEF) is observed in TC patients than in ACS patients, with a faster recovery at both discharge and follow-up [34]. As previously mentioned, the imaging diagnostic tools (mainly transthoracic/transesophageal echocardiography and cardiac magnetic resonance imaging, MRI) are crucial to highlight the apical ballooning, which is described as the apparent anomaly of the LV kinesis. This feature, together with the absence of significant obstructive lesions of the coronary arteries, confirms the initial clinical hint of a coronary syndrome with atypical onset [19].

The four modified Mayo Clinic criteria for the diagnosis include the LV regional wall motion abnormalities, the absence of a culprit coronary lesion, the onset of ECG abnormalities or the modest elevation of the cardiac troponin levels, and the absence of pheochromocytoma and myocarditis [19,20] (Figure 1).

Although cardiac MRI reveals the wall motion abnormalities and the LV ejection fraction, it is not considered the first option among the imaging techniques in the diagnostic work up of the acute phase of TC. In contrast, cardiac MRI is considered an appropriate diagnostic tool to distinguish between TC (characterized by the absence of delayed gadolinium enhancement) and myocardial infarction or myocarditis (which on the contrary exhibit distinctive patterns of enhancement) [35].

### 2.4. Pathophysiology

Stress-induced catecholamine release represents the most suggestive etiologic hypothesis for TC, which implies its initial toxic damage with a subsequent stunning of the myocardium [18,19,24]. Whether the distinctive LV wall motion abnormality of TC is triggered alternatively or in conjunction with the multivessel spasm or thrombosis, the transient occlusion of the epicardial vessels or the direct myocardial toxicity due to the stress-induced catecholamine release still needs to be investigated. The coronary endothelial dysfunction has been assumed as the pathogenetic factor unifying all of the above mentioned triggering factors [19,20,21,22,23].

### 2.5. Biomarkers

In the attempt to better describe its pathophysiology and possibly help the differential diagnosis from myocardial infarct, specific biomarkers have been proposed. Scally and colleagues showed in 55 patients with acute TC (median age, 64 years; range, 28–83 years, women = 91%) vs. 51 control subjects (median age, 63 years; range, 38–85 years, women = 90%) a higher serum concentration of IL-6 and CXCL1 chemokine (growth-regulated protein; *p* < 0.001 and *p* = 0.01, respectively). After 5 months of follow-up, an early increase in serum IL-8 concentration in TC patients (*p* = 0.07) became more pronounced (*p* = 0.009). They also reported a higher blood level of BNP, a brain natriuretic peptide normally produced in the heart and released in the event of heart stress, compared to control subjects [36]. Several investigators documented a higher level of BNP in TC, noting a marked and persistent plasma levels elevation of NT-proBNP/BNP ratio that correlated with both the extent of catecholamine increase and the severity of LV systolic dysfunction [37]. Wittstein and colleagues showed that patients with stress cardiomyopathy had supraphysiological levels of plasma catecholamines and stress-related neuropeptides whose peak values remain high, even when compared to patients with myocardial infarction [6]. Another study defined NT-proBNP as an independent predictor for short- and long-term adverse events in TC and a useful marker for risk stratification of patients [38]. Furthermore, the ratio at admission of NT-proBNP/myoglobin was employed to distinguish TC from STEMI and NSTEMI with a high sensitivity and specificity (89% and 90%, respectively, for STEMI, 65% and 90% for NSTEMI) [39]. A recent study shed light on a ST-2, a member of the interleukin-1 receptor family that acts both as a membrane receptor (ST-2) and as a secreted protein (sST-2). It is expressed by cardiomyocytes and cardiac fibroblasts as a marker of cardiac mechanical strain [40]. ST-2 was significantly elevated (*p* = 0.012) in TC patients, with an optimal cut-off of 11,018.06 pg/mL. Its increased level could be explained by cardiomyocyte strain and hemodynamic stress following acute TC. A lower sST-2 plasma level in ACS might be due to the wall motion abnormality that limits supply of the occluded coronary vessel [41,42]. Other analyzed biomarkers (troponin T, creatine kinase, creatine kinase myocardial band, C-reactive protein, H-FABP) [41,42] were not specific for TC but for other several diseases involving heart failure and the consequent state of inflammation. Overall, in accordance with the many reported data, BNP and ST-2 seem to be the most promising diagnostic biomarkers for TC.

## 3. TC and Stroke

As mentioned, the link between the two conditions looks at least bidirectional: TC has been reported as a cardioembolic source as well as a consequence of stroke. However, when secondary to acute stroke, TC can occur without emotional or psychological stress, thus suggesting a more complex and intriguing pathophysiology [7,43]. Kato et colleagues [44] proposed to categorize the clinical association between TC and stroke: (1) TC caused by central autonomic network dysfunction with cerebral infarction, (2) cardioembolic stroke caused by LV thrombus associated with TC, and (3) unknown cause and effect relation; in fact, most patients have an unclear temporal sequence relationship between TC and stroke.

### 3.1. When “Broken Heart” Causes “Broken Brain”

The cardioembolic potential of TC could perhaps be considered the most linear connection with stroke, probably mediated by the formation of LV thrombi. However, LV thrombi have a relatively low incidence among patients with TC. According to Santoro et al., only 6.6% of Takotsubo patients develop LV thrombi, and 2.2% develop a stroke [45]. In a recent study [46], the incidence of LV thrombus formation in Takotsubo patients was between 1.3 and 5.3% but the embolic risk appeared higher, with embolic strokes occurring mostly between 41 and 120 h after the onset of symptoms. Prior case series have suggested that there is a 1% to 3% risk of ischemic stroke following a TC [47]. However, this prevalence is probably higher because patients might remain asymptomatic or conversely might be unable to report any neurological impairment because of severe cardiovascular distress. A recent work evaluated risk factors for thromboembolic events in a series of 400 TC patients; among these, 8.9% of patients experienced thromboembolism, and atrial fibrillation and low LVEF on presentation were found to be associated with higher risk of thromboembolic events [48]. If we assume cardioembolism as the most likely cause of TC-provoked stroke, the above observations turn out to be extremely useful in clinical practice. It is imperative to closely monitor the neurological condition of the patient when TC is associated with reduced ejection fraction or arrhythmia.

### 3.2. When “Broken Brain” Causes “Broken Heart”

On the other hand, acute ischemic stroke may also serve as a trigger for TC [19]. Ischemic stroke can induce alterations of the cardiac function, promoting myocardial injury through different pathways. In fact, after stroke, a systemic proinflammatory response and the release of proinflammatory cytokines by damaged neuronal cells can alter sympathetic output of the hypothalamic–pituitary–adrenal axis and could lead to catecholamine release, potentially affecting cardiac function [43].

Furthermore, since the control of vasoconstriction of the coronary microcirculation is mediated by neurons that originate in the brain stem, myocardial stunning in TC due to microvascular dysfunction may be of neurogenic origin [28].

Thus, TC can originate from both functional and structural stroke-induced alterations within the central autonomic network of brain structures modulating physiological adaptation of cardiovascular function via regulation of the sympathovagal outflow to the heart. Hiestand et al. [49] demonstrated the occurrence of such structural and functional alterations within the central autonomic network with a neuroimaging study in TC patients. Increased sympathetic activity might result in activation of the renin-angiotensin–aldosterone system, which can further sustain endothelial dysfunction, increased systemic vascular resistance, and blood pressure alterations [43].

In 2008, a Japanese study [50] demonstrated that cerebral ischemic lesions on a specific anatomical brain site could be a predisposition to TC. Insular infarcts have been reported as a predominant feature of patients affected by TC [7] and can play an important role in the cardiovascular autonomic function [51]. An ischemic stroke in this area can induce systemic alterations that can in turn affect cardiac function and promote myocardial damage. Evidence supporting this crucial role of the insular cortex was collected both in clinical and preclinical models. Electrical stimulation of the rat insular cortex has been observed to trigger tachycardia, bradycardia, heart block leading to escape rhythms, and asystole [52,53]. Moreover, intraoperative insular stimulation in epileptic patients often leads to depressor responses and bradycardia [54]. Furthermore, insular ischemic lesions have been reported to determine decreased heart rate variability and an increased incidence of complex arrhythmias and sudden death [55]. Thus, damages of the insular cortex appear strictly related to alterations of the autonomic control of cardiac activity, and this may explain why acute insular ischemic lesions are so strongly associated with TC. However, extensive brainstem ischemia may also induce autonomic alterations and cause TC. In fact, the medulla also has a crucial role in the autonomic modulation of cardiovascular activity through the nucleus ambiguous, the nucleus tractus solitarius, the dorsal motor nucleus of the vagus, and the rostral ventrolateral medulla [50].

In a study of 222 patients with acute ischemic stroke [56], high concentrations of catecholamines were found to be associated with myocardial injury. It is well-known that an excessive catecholamine level constitutes a pathophysiological hallmark of Takotsubo syndrome, and, on a cardiomyocyte level, it leads to altered calcium homoeostasis and hypercontraction of sarcomeres, together with increased oxidative and metabolic stress [19,57]. This process may result in catecholamine-mediated lesions with band necrosis and impaired cardiac microcirculation [19,57]. Notably, endothelin levels in plasma are increased, and this additionally supports the hypothesis that endothelial dysfunction and microvascular constriction play an important role in the etiology of TC [58].

According to the data published by Scheitz et colleagues [43], TC has been diagnosed in 0.5–1.2% after acute ischemic stroke. Usually, transient myocardial impairment occurs within the first 10 h after a cerebrovascular event, with full or partial recovery within 3 weeks. However, as reported before, TC could be underestimated for a number of stroke patients because, in the acute phase, coronary angiography and echocardiography are not performed in emergency practice, and ECG findings are nonspecific.

These observations suggest the importance of continuous electrocardiographic monitoring in the first days of hospitalization, as well as the need for timely cardiologic evaluation in all patients with a potentially cardioembolic stroke. There is no indication at present to perform any specific blood assay for catecholamines or other cardiac enzymes in the patient with acute stroke. However, the appearance of cardiac symptoms and/or ECG changes should not be underestimated. The patient’s prognosis can vary dramatically due to diagnostic delay.

Other acute neurological disorders (i.e., seizures, head trauma, migraine, intracranial hemorrhage, subarachnoid hemorrhage, transient ischemic attack) as well as pheochromocytoma may also serve as triggers of TC [19,20]. For these reasons, it is important to keep all these particular clinical conditions in mind because the short-term outcome is poor, and the in-hospital mortality of patients with Takotsubo syndrome secondary to acute stroke is three times higher or more [43].

## 4. TC Outcome and Treatment

The prognosis of TC is typically excellent as nearly 95% of patients show a complete recovery within 4–8 weeks [21,28]. About 5% of patients who survive the acute phase have a second event, mostly occurring 3 weeks to 3.8 years after the first episode [59]. In relapses, both the morphology of the cardiac alteration (ballooning) and the trigger event may vary [19,20]; there is no clear evidence of increased susceptibility in a given clinical profile. Complications occur in 20% of cases, particularly in the early stage; they include heart failure, cardiogenic shock, onset of mitral regurgitation, mural thrombus formation, LV free-wall rupture, and arrhythmias (ventricular tachycardia, ventricular fibrillation, asystole, pulseless electrical activity, and complete atrioventricular or sinoatrial block). Death is expected between 1 and 3.2% of the cases [28]. The choice of pharmacological treatment for TC is another dilemma for cardiologists. If in the acute phase it is essential to resolve the signs of acute heart failure and to exclude an acute coronary syndrome, the greatest doubts arise in setting up a chronic pharmacological therapy. No indication exists for statins and antiplatelets drugs [20], while angiotensin-converting enzyme inhibitors or angiotensin receptor blockers are mandatory at least until the complete recovery of the function of the LV [20,24]. Beta-blockers are also indicated and may be helpful in the long term [20,23]. Whether an anticoagulant therapy is mandatory in TC remains an open issue: anticoagulation should be initiated in case of intraventricular thrombus until its dissolution and the recovery of a normal LV function [46,60].

## 5. TC Genetic Susceptibility

TC is a multifactorial and enigmatic disease with an unresolved pathogenesis. Different pathophysiologic pathways may act, individually or synergistically, but none fully explains all the processes underlying TC. Although the syndrome is not considered a primary genetic cardiomyopathy, several studies have suggested a possible genetic susceptibility in the pathogenesis of TC [61,62,63,64]. Technological advances in exome capture and DNA sequencing will shed new light on understanding the role of genetic signature in the pathogenesis of the syndrome. To date, conflicting results have been published regarding the presence or not of functional polymorphisms in candidate genes. A number of polymorphisms of adrenergic receptor or adrenoreceptor, a class of G protein-coupled receptors mediating the peripheral and central actions of catecholamines, were reported as potentially involved in the pathogenesis of TC [65]. Adrenoreceptors have been widely studied for their role in cardiac function and failure [66]. These membrane receptors exist in different subtypes related to function and tissue distribution: α1, α2, β1, β2, β3. In particular, β1 and β2 subtypes exert a key role in the regulation of excitation–contraction coupling of myocardium. The stimulation of β1 subtype results in the activation of the G stimulatory (Gs)-adenylyl cyclase (AC)-cAMP-protein kinase A, leading to an increased level of calcium and contractility. Subtype β2 has a functional role in cardiomyocyte contraction, by the activation of Gs. It also couples to pertussis toxin (PTX)-sensitive G inhibitory (Gi) proteins which inhibit AC activity and cause downstream activation of the mitogen activated protein kinases (MAPK). In addition, Giα coupling activates the cytosolic effector molecule phospholipase A2 (cPLA2), followed by cAMP independent enhancement of calcium signaling and cardiac contraction [67]. Persistent stimulation of β1AR and β2AR exhibits distinct and opposing outcomes: 1) trigger of cardiomyocyte apoptosis due to persistent β1AR activation and 2) cardioprotective effect due to persistent β2AR stimulation. Although beneficial in terms of cardiomyocyte viability, the protective effects of β2AR impair the contractile function [68]. Considering its primary role in the regulation of cardiac contraction, it is expected that polymorphisms on the adrenoreceptor could affect the physiological function of the heart [69]. A study carried out by Sharkey and colleagues [61] analyzed a group of 41 female patients vs. 43 female controls: no difference was observed in the frequency of β1AR polymorphisms compared with controls. On the contrary, a study conducted by Vriz et al. [62] for the first time highlighted a peculiar link between specific polymorphisms identified on the adrenoreceptor and the TC. In a group of 61 Caucasian patients, the authors demonstrated a significantly different distribution of Arg389 (homozygous Arg/Arg more frequent in TC) and Gln27 (homozygous Gln/Gln more frequent in healthy control), respectively, for the β1 and β2 adrenoreceptor, but no significant difference for β2AR adrenergic receptor Arg16Gly variation. These polymorphisms could be related to a dysfunction of the signaling pathway, which could explain the cardiotoxicity observed in patients with TC. Estrogen Receptor Gene 1 (ESR1) polymorphisms were associated with a greater propensity to microvascular spasm, endothelial dysfunction, and increased sympathetic stimuli in TC. A link between TC and specific ESR1 polymorphisms in postmenopausal women was observed [63]. Estrogen plays a relevant role in the release of epinephrine in presynaptic fibers [70] and in Ca++ dependent myocardial contraction [71]. In a cohort of 91 Australian patients, Figtree and colleagues recently looked for a possible correlation between TC onset and rs6915267 and rs7101752, two specific polymorphisms of ESR1 whose expression may contribute to variation in vascular reactivity in postmenopausal women. However, they did not find any association between these genetic variants of ESR1 and occurrence of the syndrome [64].

A significant correlation with TC was found by analyzing the two polymorphic loci on the ESR1 gene and specific genetic variants not yet investigated. Genotyping of ESR1 −397C>T (rs2234693); 351A>G (rs9340799)- and Estrogen Receptor Gene 2 (ESR2) −1839G>T (rs 1271572); 1082G>A (rs1256049)- genetic variants was performed in a group of 22 TC vs. 22 myocardial infarction patients and 37 healthy controls. These results underlined higher risk of experiencing TC for those study participants carrying the T allele at the rs2234693 locus of the ESR1 gene, whereas women carrying a T allele at the rs1271572 locus of the ESR2 gene demonstrated an even higher risk [63] (Figure 2).

The conflicting results in all the above mentioned studies could be due to the small size of the studied population, and further insights will be necessary to estimate the potential clinical role of the polymorphisms in TC. Undoubtedly, adrenoreceptors and ESR variants are not the only factors affecting risk of developing TC, but they constitute an important element to be considered for further studies.

Moreover, the importance of genetic signature is highlighted by a number of case reports suggesting an association between TC and rare genetic syndromes, such as CD36 Deficiency [72,73], Fragile X Syndrome [74], and Heart–Hand Syndrome [75] (Table 1).

## 6. Discussion

The diagnostic assessment of a stroke patient should always include the screening for cardioembolic sources. Several specific clinical features support a cardiac embolism as the supposed pathophysiology: maximal neurological deficit at the time of onset; syncope and/or seizure associated with the acute focal neurological deficit; multiple cerebral vascular territory involvement; lack of significant extracranial or intracranial occlusive disease other than that associated with the region of acute infarction; distinct cutoff of the affected vessel, presumably by embolus, as seen on imaging; cortical involvement in a specific vascular territory; increased risk of spontaneous hemorrhagic transformation of the infarct [76]. The most frequent cardioembolic sources are atrial fibrillation and intracardiac thrombus. Although rare, TC also has to be considered among the possible causes of embolic stroke, since its identification may prevent important consequences. However, the diagnosis of TC is often challenging. ECG abnormalities, such as ST-segment elevation, T-wave inversion, and a raise of cardiac enzyme levels may support the diagnosis, but they are not specific [3].

Moreover, cardiac enzyme and particular troponin levels may increase, but they can be found also in the early phase of acute ischemic stroke, potentially due to several underlying conditions ranging from concomitant myocardial infarction, demand ischemia, and chronic structural cardiac disease to systemic conditions. Thus, the LVEF failure with LV regional wall motion abnormalities together with the exclusion of a culprit coronary lesion, pheochromocytoma, and myocarditis seem to be the most suggestive features [19,20]. Zia et al. [77] evaluated cerebral arterial distributions of ischemic stroke associated with TC: ischemic stroke of the left middle cerebral artery was the most prevalent. According to the authors, in female patients suffering from middle cerebral artery ischemic stroke and cardiac dysfunction, TC should be considered.

Unfortunately, identifying TC in patients with acute ischemic stroke does not resolve their etiopathogenetic assessment. In fact, TC may be both a consequence and a cause of stroke [7]. For this reason, identifying the TC during the etiological research in patients hospitalized for acute ischemic stroke should not stop clinicians from continuing to look for other possible etiological causes. However, the TC can be a potential cause of mural thrombus formation in the left ventricle and, in turn, become an independent risk factor of embolic stroke causing cerebral ischemic lesions in about 1% to 3% of cases [47]. This incidence suggests that TC is an important source of embolic stroke, and, furthermore, it is probably underestimated in the acute phase because it is reversible, sometimes asymptomatic, and some stroke patients may have communication problems. Furthermore, patients have an increased rate of ischemic stroke even after the acute phase of the TC of 1.7% per patient-year. Ischemic stroke is one of the most frequent complications during the first 30 days after admission, and a mean interval between TC and cerebrovascular events of approximately one week has been reported [28,78]. Although the ischemic stroke pattern which has been typically associated to TC is cardioembolism, also other ischemic stroke subtypes should be considered while evaluating TC patients: 1) lacunar stroke, potentially complicating hypertensive fits that can be associated to stress cardiomyopathy, and 2) watershed infarcts, as expression of an acute hypoperfusion that could be due to a massive LVEF failure in some TC cases [22,79].

For all these reasons, stroke physicians are always recommended to search for this condition in all their patients, particularly in embolic strokes that affect elderly females with insular infarcts, in order to promptly detect and manage it. Thus, in the acute phase of the management, ECG, long-term ECG monitoring and repeated ultrasound examinations together with ischemic myocardial markers, including troponin-T, creatine kinase, and BNP as well as NT-proBNP levels, need to be performed in all stroke cases. So far, many other biomarkers have been studied and proposed. Acutely, IL-6 and CXCL1 chemokine have been shown to increase in TC patients, and, after 5 months of follow-up, an early increase in serum IL-8 concentration has been reported [40]. In a recent study, ST-2 has been proposed as a new possible biomarker for the differential diagnosis of acute coronary syndrome from TC because it was significantly elevated in TC compared to ACS patients [41,42].

Currently, several observations have hypothesized a putative genetic susceptibility for TC. Both familial and recurrent cases of TC have been described, suggesting a possible influence of the genetic background in the pathogenesis of the syndrome. A genetic predisposition has been suggested based on the few familial TC cases. Although a clear link between TC and genetic factors has not yet been identified, several polymorphisms have been proposed. Two principal studies focused on adrenergic receptor and ESR genes [3], encoding for the main proteins involved in physiological cardiac function, although conflicting results did not lead to a well-defined picture of the TC pathophysiology. Interestingly, a very recent systematic review has been performed to cast light on the potential genetic and epigenetic factors significantly associated with TC [80]. The data obtained suggested that genetic variants of ADRB1, GRK5, and BAG3 genes, or variants of APOE, MFGE8, ALB, APOB, SAA1, A2M, and C3 genes, can probably interact with the environment and emotional/physical stress and predispose certain subjects more than others to develop TC. This might justify the heterogeneity of TC acute episodes, with clinical pictures accompanied by more severe, or not, complications. Although environmental triggers and concomitant comorbidities are pivotal in TC development [81], the genetic heterogeneity and a potential polygenic predisposition may also play a contributory role mainly by determining the dysregulation of the adrenergic system [82].

## 7. Future Perspectives

More sophisticated molecular studies by using novel technologies might lead to further and larger mechanistic studies both in animals and human cell models for identifying specific pathways and for clarifying the complex TC pathophysiology. It would be desirable to establish a combination of biomarkers that could be useful for the clinicians to distinguish TC from other conditions. Such molecular panels should be utilized in conjunction with other clinical investigations, thus giving a useful contribution for the identification of candidates for long-term therapies, which are at increased risk, and helping in follow-up studies (Figure 3).

Peculiar triggers for TC should be investigated while exploring medical history, such as recent delivery or surgical procedures. Recently TC has been also described as a possible consequence of the cytokine storm secondary to COVID-19 infection, and some rare cases of TC were also reported following immunization with mRNA COVID-19 vaccines [83,84]. Thus, clinicians need to carefully follow these patients during hospitalization and after their discharge in order to reduce the rate of cerebrovascular events. Abanador-Kamper et colleagues suggest that apical ballooning with severe anterior wall motion abnormalities followed by a rapid improvement in left ventricular morphology and function may facilitate the formation of cardiac embolism and consequently increase stroke rates in TC [85]. It is of special interest whether medical treatment influences the rate of developing an ischemic stroke after the diagnosis of TC. Actually, no therapeutic guidelines exist, and TC patients are usually treated with antithrombotic and heart failure therapy with a low bleeding risk and a high short-term survival [86]. However, literature results point to a potential role for oral anticoagulation in high-risk patients until recovery of left ventricular function [85]. Indeed, Santoro et al. proposed a therapeutic algorithm for oral anticoagulation [87] with which oral anticoagulation was considered only for high-risk patients with apical ballooning pattern and increased admission levels of troponin-I until recovery of left ventricular function. However, despite these results, the benefit of anticoagulation as well as the choice among anticoagulants are still not established; randomized controlled trials assessing the role of anticoagulation in TC patients preventing an ischemic stroke are therefore highly needed.

## Figures and Tables

**Figure 1 jpm-12-01244-f001:**
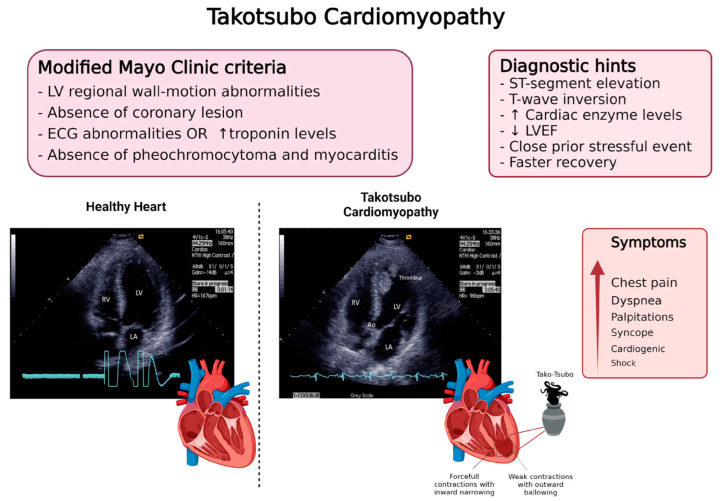
A schematic representation of the main parameters for the diagnosis of Takotsubo Cardiomyopathy. Transthoracic echocardiography in TC, here compared to a healthy pattern, shows left ventricle (LV) with akinetic apex and thrombus. LA = left atrium; RV = right ventricle; Ao = aorta.

**Figure 2 jpm-12-01244-f002:**
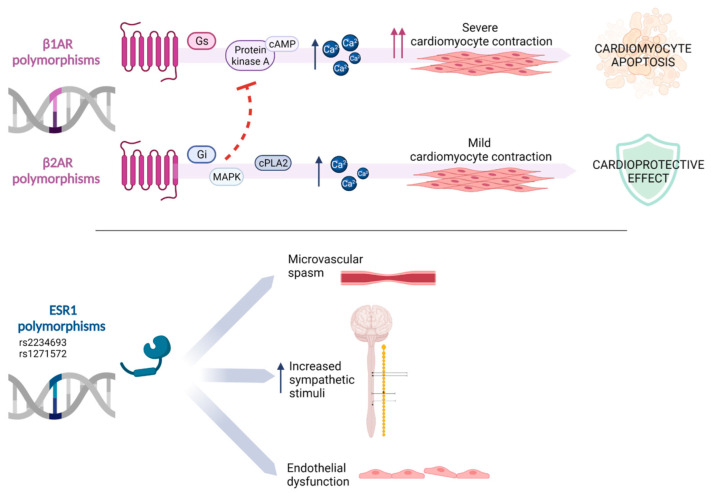
Genetic susceptibility in the pathogenesis of TC: Adrenergic Receptor β1, β2 (ARβ1, ARβ2), and Estrogen Receptor Gene 2 (ESR2) polymorphisms could explain the cardiotoxicity observed in TC patients. Stimulatory G protein (Gs), cyclic Adenosine MonoPhosphate (cAMP), mitogen-activated protein kinase (MAPK), cytosolic phospholipase A2 (cPLA2). Created with BioRender.com.

**Figure 3 jpm-12-01244-f003:**
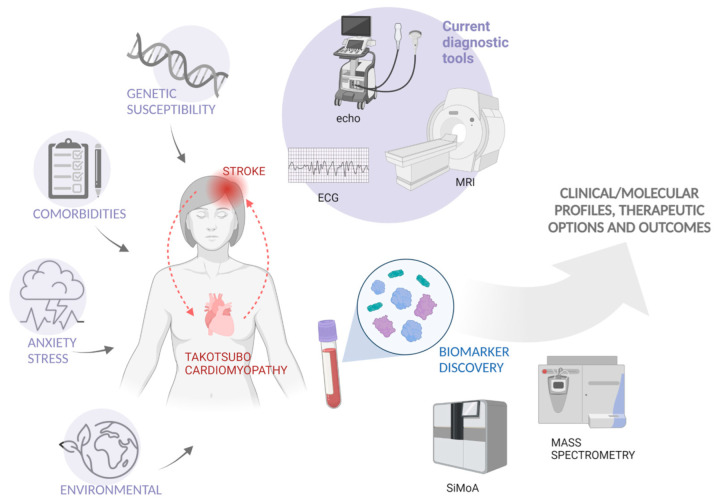
A schematic representation of the association between Takotsubo cardiomyopathy (TC) and acute ischemic stroke, as a result of putative genetic predisposition, environmental triggers, and concomitant comorbidities. The biomarker discovery by innovative technologies could be useful to investigate clinical profile, therapeutic options, and outcomes of TC patients. SiMoA (Single Molecule Array). Created with BioRender.com.

**Table 1 jpm-12-01244-t001:** Case reports suggesting an association between TC and rare genetic syndromes.

Genetic Syndromes	Age	Sex	Clinical Picture and Hypotheses about the Causes	Ref.
CD36 deficiency	71	F	The patient reported chest pain and worsening of dyspnea and during the investigations type I CD36 deficiency was diagnosed. DNA sequencing showed that the patient had a compound heterozygosity of the CD36 gene (a nucleotide change in C478T and an adenine insertion at nucleotide 1159 in exon 10)	[72]
CD36 deficiency	77	F	No genetic studies have been carried out	[73]
Fragile X-Syndrome	67	F	Genetic testing revealed that the subject was a carrier of the premutation condition, 80 CGG repeats. The patient’s second grandson, in fact, was a carrier of the complete mutation of FMR1 gene (350–400 repetitions) and was depicted by the classic traits of fragile X syndrome. This case report suggests an association of TC with a mutation of the gene encoding the fragile X-syndrome. The expression of expanded FMR1 allele with CCG repeats on the X-chromosome in a conductor of fragile X syndrome may explain the preferential occurrence of TC in women	[74]
Heart-Hand Syndrome	60	F	The patient reported chest pain that occurred after a quarrel. After investigation she received a diagnosis of TC and a bilateral IV toe brachydactyly was noted. Across medical literature, many described syndromes are characterized by an association between skeletal alteration of the hands and heart abnormalities (heart-hand syndrome). The authors wonder if the case can belong to this kind of complex malformations as well as if a molecular-genetic trait may have a role for two clinical aspects	[75]

## Data Availability

I.C. and A.B. take responsibility for the integrity of the data and the accuracy of the data analysis.

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
