# Peer review of "The Octopus Trap of Takotsubo and Stroke: Genetics, Biomarkers and Clinical Management"

_jpm, 2022, doi:10.3390/jpm12081244_

Round 1

Reviewer 1 Report

1.Title and abstract

The title and abstract is consistent with the presented problem and reflects the main message of the study.

2. Introduction

Introduction is clear and helpful to readers unfamiliar with the subject.

3. Discussion
The interpretation of the results are clearly presented and adequately supported by the evidence adduced.

4. Conclusions
The conclusions are logically valid.

5. Graphics

All figures are adequate and necessary.

Author Response

Reviewer #1 

  1. Title and abstract. The title and abstract is consistent with the presented problem and reflects the main message of the study.
  2. Introduction. Introduction is clear and helpful to readers unfamiliar with the subject.
  3. Discussion. The interpretation of the results are clearly presented and adequately supported by the evidence adduced.
  4. Conclusions. The conclusions are logically valid.
  5. Graphics. All figures are adequate and necessary.

Authors' Reply to Reviewer #1:

We sincerely thank the Reviewer for the kind comments on our work. We are truly glad that he/she appreciated the manuscript in content and form. The manuscript has now been further updated according to other Reviewers' suggestions, and we hope that also this new version would satisfy R#1's worthy opinion.

Reviewer 2 Report

First of all, I would like to thank the authors due to the choice to perform a review about Takotsubo and its relation with one of the main neurological conditions, stroke. As clinical neurologist I consider it a very relevant topic. 

But I do have some considerations prior to evaluate the possibility of the publication of the paper; they are mainly about the structure of the work itself: 

1. In supplementary material add a flowchart about Research Strategy. I am aware that is not a systematic review but I think it could be very interesting. It gives information about quality of the research done. 

2. Explaining Takotsubo prevalence in other neurological conditions such as after status epilepticus could be of interest. Compared to other clinical context Takotsubo in context of stroke has it better/worser or same prognosis or clinical course? I think that information should be added. 

3. Abstract: Remove last sentence o rephrase it, it doesn't add information. Line 23: remove extra ".". 

4. Cardiological assessment of Takotsubo reorganize it if you want to be it really useful and attractive for clinicians. I would suggest this structure: risk factors; symptoms; diagnosis tests (ECG;  ECO; blood sample -not only the amount of cardiac enzymes-try to specify the magnitude of the difference- , it is not a difference in the velocity of the increase of these enzymes compared to acute coronary disease?-; other test: Angio-TC, Angio-RM and coronography -angiography is always performed? In my clinical experience varies according to the clinical context and the results of other test, specify it); and etiology section (talk about possible pathogenic mechanisms). 

5. Reorganize for the same reasons the structure of the section TC and stroke: First add information about TC as risk factor of stroke (Prevalence, possible mechanisms and clinical tips about when to suspect) and secondly TC in acute of subacute phase of stroke (Prevalence, possible mechanisms and clinical tips about when to suspect). Using some kind of panel or table could be useful. 

Adding a consideration about the possible influence of TC at the moment of the treatment of stroke could be of interest. Explain if it can modify the acute treatment or the subacute treatment of stroke itself (can influence in the decission and how?). 

6. TC outcome and treatment: Explain better the clinical profile of the recurrence. Are there risk factors that increase the probability of recurrence? The recurrence occurs in the first month/year...? The severity of a recurrent episode is the same?

7. Genetic Features: I do believe that the section could be reduced significantly without losing information. For instance, the clinical cases mentions could go perfectly in a table. 

8. Other biomarkers section: I am not sure if the evidence about them is sufficient to have a separate section. Incorporate after current diagnostic test section or create a section about future research lines and add in this section information about effort done to improve clinical care of this subjects (treatment aspects). 

9. Figure 3: I would avoid interrogation symbol; stress could ve misunderstood it is referring to stress induced by organic conditions or induced by anxiety? I think that in this figure a summary of current diagnostic tools used in clinical practice could be useful and not only possible future diagnostic or prognostic tools. 

I would not continue with further comments before the authors answer to all these concerns. 

Author Response

Reviewer #2

First of all, I would like to thank the authors due to the choice to perform a review about Takotsubo and its relation with one of the main neurological conditions, stroke. As clinical neurologist I consider it a very relevant topic. But I do have some considerations prior to evaluate the possibility of the publication of the paper; they are mainly about the structure of the work itself:

  1. In supplementary material add a flowchart about Research Strategy. I am aware that is not a systematic review but I think it could be very interesting. It gives information about quality of the research done.
  2. Explaining Takotsubo prevalence in other neurological conditions such as after status epilepticus could be of interest. Compared to other clinical context Takotsubo in context of stroke has it better/worser or same prognosis or clinical course? I think that information should be added. 
  3. Abstract: Remove last sentence o rephrase it, it doesn't add information. Line 23: remove extra ".". 
  4. Cardiological assessment of Takotsubo reorganize it if you want to be it really useful and attractive for clinicians. I would suggest this structure: risk factors; symptoms; diagnosis tests (ECG;  ECO; blood sample -not only the amount of cardiac enzymes-try to specify the magnitude of the difference- , it is not a difference in the velocity of the increase of these enzymes compared to acute coronary disease?-; other test: Angio-TC, Angio-RM and coronography -angiography is always performed? In my clinical experience varies according to the clinical context and the results of other test, specify it); and etiology section (talk about possible pathogenic mechanisms).
  5. Reorganize for the same reasons the structure of the section TC and stroke: First add information about TC as risk factor of stroke (Prevalence, possible mechanisms and clinical tips about when to suspect) and secondly TC in acute of subacute phase of stroke (Prevalence, possible mechanisms and clinical tips about when to suspect). Using some kind of panel or table could be useful. Adding a consideration about the possible influence of TC at the moment of the treatment of stroke could be of interest. Explain if it can modify the acute treatment or the subacute treatment of stroke itself (can influence in the decission and how?).
  6. TC outcome and treatment: Explain better the clinical profile of the recurrence. Are there risk factors that increase the probability of recurrence? The recurrence occurs in the first month/year...? The severity of a recurrent episode is the same?
  7. Genetic Features: I do believe that the section could be reduced significantly without losing information. For instance, the clinical cases mentions could go perfectly in a table. 
  8. Other biomarkers section: I am not sure if the evidence about them is sufficient to have a separate section. Incorporate after current diagnostic test section or create a section about future research lines and add in this section information about effort done to improve clinical care of this subjects (treatment aspects). 
  9. Figure 3: I would avoid interrogation symbol; stress could ve misunderstood it is referring to stress induced by organic conditions or induced by anxiety? I think that in this figure a summary of current diagnostic tools used in clinical practice could be useful and not only possible future diagnostic or prognostic tools.

I would not continue with further comments before the authors answer to all these concerns.

Authors' Reply to Reviewer #2

We thank the Reviewer for this accurate and expert revision. Here we provide a point-by-point reply to his/her requests.

  1. Thanks for this useful suggestions. As advised, we have included a description of our research strategy in the "Supplementary Materials" section.
  2. Thank you for this stimulating input. We have supplemented the manuscript in paragraph “Introduction” with what you suggested.  
  3. Thank you for the advice. We have edited the manuscript according to your suggestion both in abstract and in line 23.
  4. Thank you for the advice. A reorganization of this chapter has been made according to your suggestions
  5. Thank you for this stimulating and appropriate advice. We have modified the "CT and Stroke" paragraph by reorganizing it and adding some more clinically oriented content 
  6. Thank you for the suggestion. We have integrated this into the manuscript in the section "CT outcome and treatment”
  7. Thank you, we have organized the text as suggested in a summary table
  8. Thank you; as you suggested, we have incorporated, as well as little reduced, the biomarker section after the diagnostics section.
  9. Thank you for the suggestion. Figure 3 has been modified according to your suggestion

We have appreciated the thorough evaluation of the original manuscript and the opportunity to improve its quality by these many useful suggestions, that we have followed in the revision process. We sincerely hope that the Reviewer will now find adequately improved the paper, and we are sure open to other suggestions if needed. 

Reviewer 3 Report

Canavero et al are presenting a review on  genetics,  biomarkers and clinical management on Takosubo and its relation to stroke. The manuscript is very well writeen and updated. I just would like to see a few sentences on "future research priorities" / unsolved questions on Takosubo and stroke.

Kind regards

Author Response

Reviewer #3

Canavero et al are presenting a review on genetics,  biomarkers and clinical management on Takosubo and its relation to stroke. The manuscript is very well writeen and updated. I just would like to see a few sentences on "future research priorities" / unsolved questions on Takosubo and stroke. Kind regards

Authors' Reply to Reviewer #3:

Many thanks for the positive comment toward the work. We agree with the reviewer's suggestion and, in order to make it easier to read, we have enclosed a new paragraph on future perspectives in research, that actually helps to conclude the whole review. We hope that this part will satisfy the reader's curiosity about unresolved issues and possible research perspectives. We have also made other changes to the first version of the manuscript according to other Reviewers' advice, and we sincerely hope R #3 will find our work still interesting and worthy for publication.